# Transcriptional Landscape of 3D vs. 2D Ovarian Cancer Cell Models

**DOI:** 10.3390/cancers15133350

**Published:** 2023-06-26

**Authors:** Rachel Kerslake, Birhanu Belay, Suzana Panfilov, Marcia Hall, Ioannis Kyrou, Harpal S. Randeva, Jari Hyttinen, Emmanouil Karteris, Cristina Sisu

**Affiliations:** 1Division of Biosciences, College of Health, Medicine and Life Sciences, Brunel University London, Uxbridge UB8 3PH, UK; 2Computational Biophysics and Imaging Group, The Faculty of Medicine and Health Technology, Tampere University, 33100 Tampere, Finland; 3Mount Vernon Cancer Centre, Rickmansworth Road, Northwood HA6 2RN, UK; 4Warwickshire Institute for the Study of Diabetes, Endocrinology and Metabolism (WISDEM), University Hospitals Coventry and Warwickshire NHS Trust, Coventry CV2 2DX, UK; 5Warwick Medical School, University of Warwick, Coventry CV4 7AL, UK; 6Research Institute for Health & Wellbeing, Coventry University, Coventry CV1 5FB, UK; 7Aston Medical School, College of Health and Life Sciences, Aston University, Birmingham B4 7ET, UK; 8Laboratory of Dietetics and Quality of Life, Department of Food Science and Human Nutrition, School of Food and Nutritional Sciences, Agricultural University of Athens, 11855 Athens, Greece

**Keywords:** ovarian cancer, high-grade serous ovarian cancer (HGSOC), monolayer, 2D, 3D, scaffold, tumour microenvironment (TME), extracellular matrix (ECM), collagen, Matrigel, agarose

## Abstract

**Simple Summary:**

Ovarian cancer is one of the most lethal female cancers. Numerous investigations into the development and progression of this disease have resulted in the creation of numerous three-dimensional culture models to better reflect the natural microenvironment of these tumours. In this study, we leverage the available transcriptomics and clinical and novel experimental data to evaluate the impact of the growth conditions on various cancer cells and examine whether they better approximate the behaviour of tumour cells compared to the classical two-dimensional models. Our results show that variability in the growth conditions can impact key genes and biological processes that are hallmarks of cancer, highlighting the need for future studies to identify which is the most appropriate in vitro/preclinical model to study tumour microenvironments.

**Abstract:**

Three-dimensional (3D) cancer models are revolutionising research, allowing for the recapitulation of an in vivo-like response through the use of an in vitro system, which is more complex and physiologically relevant than traditional monolayer cultures. Cancers such as ovarian (OvCa) are prone to developing resistance, are often lethal, and stand to benefit greatly from the enhanced modelling emulated by 3D cultures. However, the current models often fall short of the predicted response, where reproducibility is limited owing to the lack of standardised methodology and established protocols. This meta-analysis aims to assess the current scope of 3D OvCa models and the differences in the genetic profiles presented by a vast array of 3D cultures. An analysis of the literature (Pubmed.gov) spanning 2012–2022 was used to identify studies with paired data of 3D and 2D monolayer counterparts in addition to RNA sequencing and microarray data. From the data, 19 cell lines were found to show differential regulation in their gene expression profiles depending on the bio-scaffold (i.e., agarose, collagen, or Matrigel) compared to 2D cell cultures. The top genes differentially expressed in 2D vs. 3D included C3, CXCL1, 2, and 8, IL1B, SLP1, FN1, IL6, DDIT4, PI3, LAMC2, CCL20, MMP1, IFI27, CFB, and ANGPTL4. The top enriched gene sets for 2D vs. 3D included IFN-α and IFN-γ response, TNF-α signalling, IL-6-JAK-STAT3 signalling, angiogenesis, hedgehog signalling, apoptosis, epithelial–mesenchymal transition, hypoxia, and inflammatory response. Our transversal comparison of numerous scaffolds allowed us to highlight the variability that can be induced by these scaffolds in the transcriptional landscape and identify key genes and biological processes that are hallmarks of cancer cells grown in 3D cultures. Future studies are needed to identify which is the most appropriate in vitro/preclinical model to study tumour microenvironments.

## 1. Introduction

Ovarian cancer (OvCa) is one of the most lethal gynaecological malignancies of the 21st century. Affecting over 313,000 women worldwide, OvCa typically presents at a late stage with non-specific symptoms, causing a detriment to survival outcomes, which fall as low as 20% [1]. The metabolic processes involved in OvCa aetiology, however, remain poorly understood. There are three main histological types of OvCa. Epithelial OvCa accounts for 90% of all cases, with high-grade serous ovarian cancer (HGSOC-70%) being the most prevalent of the five subtypes as well as the most lethal [1]. Other subtypes include low-grade serous ovarian cancer (LGSOC-5%), endometrioid adenocarcinoma of the ovary (EAC-10%), clear cell carcinoma (CCC-10%), and mucinous adenocarcinoma (MAC < 3%). The least common are germ-line and stromal sex cord tumours, which cover 10% of cases [2].

In order to gain a better understanding of the events that take place within the tumour microenvironment (TME), a model capable of emulating the in vivo milieu is required. The use of conventional monolayer cell cultures (two-dimensional; 2D) allows for analysis using a controlled in vitro environment to investigate the physiological, morphological, and biochemical properties of biological systems [3]. Monolayer cultures have served as an integral foundation of biological research since the introduction of immortalised HeLa in 1951, paving the way for thousands of subsequent cell lines [4]. Cell models have since proven invaluable in the modelling of normal physiology and diseases, including cancer [5].

Nevertheless, monolayer cultures have translational limitations, with differences in gene expression, drug response, and cell signalling evident when compared to in vivo models [6]. Many processes related to tumorigenesis and metastasis are often over-simplified in monocultures [7]. As a result, monolayer cultures often fail to recapitulate the complex microenvironment, diffusion gradients, and cellular characteristics associated with in vivo systems, thus leading to variation from the predicted response in animal and computational modelling, as well as clinical testing [6,8].

As global research efforts strive to answer increasingly complex biological questions, there is a greater need for a representative system capable of physiological emulation. Many studies have shown that the complexities of tissue organisation, differentiation, and gene expression are demonstrated at higher levels in three-dimensional (3D) cell cultures [9,10]. This setup allows for cells to be grown in an environment that sustains spatial complexities representative of in vivo conditions, allowing cells to differentiate and interact in a tissue-specific manner [11]. The key differences between monolayer and 3D cultures are summarised in Table 1 [5,12].

Further evidence emphasises the importance of the TME for maintaining cancer stemness, exerting a significant effect on gene expression [13]. The integration of an extracellular matrix (ECM) i.e., a scaffold, provides the necessary environment for this 3D cellular growth and differentiation [14]. Scaffolds emulate the tissue–tissue interfaces and chemical gradients required within a living system. Recent advancements include 3D organoid systems capable of sustaining a vast array of tumour models, including glioblastoma, colon, lung, and ovarian [15,16,17].

Epithelial OvCa cells grown in 3D often present histological features characteristic of the original tumour in situ [18]. Three-dimensional epithelial OvCa cell lines also presenting with a reduced proliferative rate are thought to be enabled by a synthetic ECM [19]. An enhanced response to external stimuli is also evident in OvCa cultures. Thus far, 3D OvCa cultures have proven particularly useful as a model of therapeutic resistance, capturing developed resistance to platinum-based therapeutics similar to an in vivo OvCa response. The OvCa cell line SKOV-3, for example, demonstrates a higher degree of chemoresistance to both cisplatin and paclitaxel when cultured in 3D [20]. Moreover, colorectal and pancreatic cancer cells grown in 3D exhibit differential gene expression, which is associated with augmented ATP production in 3D cultures. Subsequently, the amino acid production and metabolomic activity of glycolytic intermediates are increased when compared to monolayer substrates of the same cell line [21,22].

A wide array of scaffolds can be used to recapitulate the TME and support the differentiation of 3D cultures, given that the TME is pivotal for the regulation of a diverse array of processes, including migration, proliferation, differentiation, and cell–cell communication [23]. Often interchangeable in the literature, spheroids and organoids differ in complexity. Typically, spheroids are rounded and comprise cells grown initially in 2D, and as such, retain some simplicity of gene expression. Growth is often achieved using the hanging drop method or an ultra-low attachment plate and is ideal for the study of diffusion gradients and core hypoxia [24].

Given the current trajectory of 3D cancer models and their appeal to support the reduction of animal research, it is therefore safe to assume that a complex OvCa on a chip model will soon be achievable. This meta-analysis aims to evaluate the current landscape of OvCa cell models to elucidate differences presented in their genetic profile and associated signalling pathways when grown in 3D compared to 2D monolayer cultures, using published RNAseq and microarray datasets.

## 2. Materials and Methods

### 2.1. Study Design

The data mining process was designed with the intent to search the current literature for studies modelling OvCa using 3D culture techniques alongside a 2D control and assess the differences in gene regulation between 2D and 3D cultures. The National Centre for Biotechnology Information (NCBI) PubMed database was searched for studies relevant to the scope of this work between the years 2012 and 2022 (Figure 1). No limitations to the original language were applied as long as English translations were available. The filter for human studies was utilised. The search terms applied included: “cancer” AND “ovar*” AND “3d” NOT “sound” NOT “ultra” NOT “imaging” NOT “ultrasound” NOT “review”. The literature that was inaccessible via the university institutional access was also removed. Additional searches through the NCBI, Sequence Read Archive (SRA), and Gene Expression Omnibus (GEO) accession platforms were also utilised.

Inclusion criteria: Studies were included if they encompassed 3D OvCa models as well as 2D comparisons. In addition, those with associated data from sequencing arrays and RNA sequencing accessible through GEO or SRA were also sought.

Exclusion criteria: Studies were discarded if they did not meet the original search criteria. Additional studies that were excluded comprised those with a lack of comparative 2D cultures, no open access, and no human samples, i.e., the use of animal (usually murine) cell lines. The final exclusion criteria for enrichment encompassed studies with no associated data. 

### 2.2. Cell Cultures and 3D Modelling

Unless otherwise stated, all reagents were purchased from Thermo Fisher Scientific. The serous ovarian adenocarcinoma cancer cell line SKOV-3 (European Collection of Authenticated Cell Cultures (ECACC 91091004), Salisbury, United Kingdom) were seeded in conventional culture-treated polystyrene T75 flasks. The cells were grown in Dulbecco’s Modified Eagle’s Medium (DMEM) supplemented with 10% foetal bovine serum and 1% penicillin–streptomycin. The media were changed every 2–3 days, with experimental work proceeding after 3 passages. The cell suspension concentrations were calculated using the trypan blue exclusion method. For monolayer substrate comparison, the cells were seeded in triplicate at a density of 1.6 × 10^6^ cells/mL in an Ibidi 8-well chamber (Ibidi, Munich, Germany) with complete medium. Three-dimensional cultures were generated using a 1:12 ratio of cells suspended in medium mixed with GelTrex^TM^ (batch: 2158356). Each well contained a final concentration of 300 μL. The chamber was left to incubate at 37 °C for 30 min to allow for gelation. Then, 100 μL of media was added to each well. Media changes took place every 2–3 days up to day 10. Images were captured each day using a Nikon TS100 Inverted Phase Contrast light microscope (Nikon, Tokyo, Japan).

A certificate of analysis and a declaration of mycoplasma-free cultures were provided upon receipt of the cells from the ECACC and validated in-house with DAPI staining; the cells were used following 3 passages from purchase.

### 2.3. Immunofluorescent Imaging

On day 10, the media was removed. Both the 2D and 3D cultures were fixed with 4% paraformaldehyde in PBS for 10 and 30 min, respectively. The chambers were washed ×3 with PBS following incubation with 0.1% triton-x for 10 min. The chambers were again washed prior to blocking with 10% bovine serum albumin (BSA) (Sigma Aldrich, Burlington, MA, USA) for 1 h at room temperature. The BSA was then removed for phalloidin (ATTO-TEC, Siegen, Germany) actin staining, using a 1:1000 dilution in 1% BSA for 30 min at room temperature. The chambers were again washed ×3 with PBS before the administration of a final DAPI (Invitrogen, Waltham, MA, USA) nuclear stain for 10 min. The samples were washed to remove residual DAPI and then kept hydrated in PBS prior to imaging.

### 2.4. Laser Scanning Confocal Microscopy

Laser scanning confocal microscopy (LSM780, Carl Zeiss, Oberkochen, Germany) was used for the 3D imaging of cells cultured in a glass substrate and encapsulated in 3D Geltrex hydrogel. The cell samples were subject to excitation/emission wavelengths at 405 nm/410 nm–495 nm and 488 nm/495 nm–620 nm for imaging of the nuclei (DAPI) and actin (phalloidin), respectively. The emitted fluorescence signal was recorded using photomultiplier tube (PMT) detectors. The optical Z-stacks were acquired using a 63× objective (A plan-Apochromat 63×/1.4 Oil immersion, Carl Zeiss, Oberkochen, Germany). The laser power, detector gain, and scan speed were optimised to avoid photobleaching. The image size was 2048 pixels × 2048 pixels, with a voxel size of 40 nm × 40 nm in the XY-plane, and 250 nm in the Z-direction. The images were deconvoluted using the automatic deconvolution mode with the theoretical point spread function in Huygens Essential software (Scientific Volume Imaging, Hilversum, The Netherlands). Avizo software (Thermo Fisher Scientific, Waltham, MA, USA) was used for 3D visualisation.

### 2.5. RNA Sequencing-Sequence Read Archive (SRA)

NIH Sequence Read Archive (SRA) data were found using the same search terms outlined in the study design. SRA data, in the form of RNA sequencing reads produced by Illumina NextSeq 500 and Illumina HiSeq 2500, were acquired for re-analysis, and the accession IDs are outlined below in Table 2. Briefly, the relevant data, in the form of FASTQ files, were transferred from the SRA database via Amazon Web Services for in-house analysis (Table 2)-the full list can be seen in Appendix A. The corresponding scaffolds used in each study were as follows. PRJNA472611 3D cells were embedded in agarose; PRJNA564843 cells were grown upon a layer of omental fibroblasts embedded in collagen; PRJNA530150 3D cells were grown in Matrigel.

The raw RNAseq data were produced using the method previously described to standardise the results for comparison [25]. Briefly, TopHat2 (v.2.1.1) was applied to align reads to the reference human genome GRCH38 using the ultra-high-throughput short-read aligner Bowtie2 (v.2.2.6). Using Samtools (v.0.1.19), applicable replicates were merged according to a selection criterion, taking only high-quality mapped reads (<30). Subsequent transcript assembly and quantification followed using Cufflinks (v.2.2.1). Finally, differential expression profiles were obtained for further analysis using Cuffdiff (v.2.2.1). The false discovery rate (FDR) used for the analysis was set at 0.2.

### 2.6. RNA Sequencing-Statistical Analysis

The expression data were analysed in R (v.4.1.0, The R Foundation for Statistical Computing, Vienna, Austria) with the R Studio desktop application (v.2022.07.2, RStudio, Boston, MA, USA), using specific libraries for modelling, visualisation, and statistical analyses for the identification of differentially expressed genes (DEGs). Similar to our previous work, the Pearson correlation coefficient was applied for the estimation of gene expression patterns, and Student’s t-test was utilised to assess statistical significance between the expression profiles (i.e., 2D vs. 3D). Significance thresholds were set for a *p*-value < 0.05. For the identification of enriched pathways in omics data, pathfindR was employed. For visualisation, volcano plots were generated using R package ggplot2 (v.3.3.5). DEGs were identified and isolated for subsequent enrichment analysis. Furthermore, we used the OmicsPlayground (v2.8.14, BigOmics Analytics, Bellinzona, Switzerland) online application for exploring the transcriptional landscape of ovarian cancer cells grown in 2D and various 3D systems using as scaffolds agarose, collagen, and Matrigel [26].

### 2.7. Gene Expression Omnibus (GEO) Array-Statistical Analysis

Genomic datasets (accession numbers: PRJNA232817 and PRJNA318768) were downloaded from the NCBI public repository GEO archive. These OvCa cells were grown using the ultra-low attachment and hanging drop techniques. The GEO2R web application was accessed to re-analyse the expression data in line with the research questions in this study (control 2D samples vs. control 3D samples). The thresholds were again set at a *p*-value < 0.05, FDR = 0.2, and Log_2_FC > 1, applying the Benjamini and Hochberg procedure for decreasing the false discovery rate. Volcano plots were generated using GEO2R (https://www.ncbi.nlm.nih.gov/geo/geo2r/ (accessed on 1 August 2022)).

### 2.8. Functional Enrichment Analysis

The differentially expressed genes (DEGs), identified through GEO2R and SRA analysis, were then subjected to functional enrichment analysis. Funrich (v.3.1.3) was accessed to provide a functional annotation, including associated sites of expression, biological processes, and pathways. Enrichment analysis was performed using Omics Playground for the functional comparison of the OvCa genes in 2D vs. 3D [26].

### 2.9. Presentation of Data and Statistical Analysis

Global distribution infographics were generated using R (v.4.1.0) in R Studio (v.2022.07.2) along with ggplot2 (v.3.3.6), maps (v.3.4.0), and world map data from Natural Earth (0.1.0). Subsequent comprehensive background analysis and graphs pertaining to the publication data, cell-line frequency, and associated characteristics were generated using GraphPad Prism9^®^ (v.9.4.1-GraphPad Software, Inc., San Diego, CA, USA). The statistical reliability of the Omics Playground data was ensured through the incorporation of Spearman’s rank correlation, GSVA, ssGSEA, GSEA, and the Fisher exact test.

## 3. Results

### 3.1. Three-Dimensional Ovarian Cancer Models

#### 3.1.1. Literature Overview

The geographical spread of the fifty studies selected suggests that the United States of America (USA) is the top publisher of 3D OvCa modelling, with over 50% of the research accessed originating from the USA. China, Italy, Korea, and the UK follow, with the majority of the work originating from Europe or North America (Figure 2A,B).

To achieve 3D cultures, cell lines are grown in a fabricated ECM, also known as a scaffold. In the literature, the most commonly used scaffolds for 3D OvCa growth are pre-coated low-attachment plates, followed by Matrigel, the hanging drop method, and plant-based hydrogel (Figure 2F). Over 43 unique OvCa cell lines have been utilised throughout the studies (Figure 2C). The top 10 represent an array of OvCa subtypes (Figure 2D). The ovarian carcinoma cell line SKOV-3 is the most frequent in the literature, appearing in 19 instances. The trend of studies focusing on OvCa subtypes was compared with the actual global incidence rates. For epithelial OvCa, the cell models used follow a similar trend of frequency to actual global incidences, with HGSOC being the most prevalent form of OvCa and also the most studied. Of note, stromal sex cord and granulosa OvCa comprise 10% of global cases; however, no 3D models were found in the studied literature. The genomic ancestry of the cell lines is often overlooked; however, given the disparity in care, the backgrounds of the cell lines used were also sourced (Figure 2E). A disproportionate number of cell lines used are either White (N = 80) in origin or are considered unclassified i.e., there are no available data (N = 30).

#### 3.1.2. Differentially Expressed Genes

The data accessed through SRA and GEO were screened for OvCa cells grown in 2D and 3D under similar conditions. Three separate studies were chosen, encompassing 19 cellular models grown under normal conditions in agarose, Matrigel, and collagen-based scaffolds. All cell lines grown in 3D showed differential gene expression when contrasted with the same cell lines under the same conditions but grown in 2D (Figure 3). The number of statistically significant differentially expressed genes (DEGs) with *p* < 0.05 between the 2D and 3D cultures ranged between 234 in PEO1 and 1429 in the OVCAR5 cell line.

The HGSOC OVCAR8 appeared in all three studies with different accompanying scaffolds: Matrigel, agarose, and collagen. Additional analysis explored the effects of the different scaffolds on the genetic profile of these cells (Figure 4). All conditions influenced the differential regulation of OVCAR8’s transcriptional profile. A total of 13 DEGs were identified (Table 3) based on their common dysregulation among scaffolds when grown in 3D. Similarly, these genes were seen to feature highly in the other 3D models, e.g., the dysregulation of *ANGPTL4* appeared in 12/19 of the studies. When comparing the DEGs identified in the OVCAR8 cells grown in 2D and 3D, eight were found to be common regardless of their scaffold type (Figure 4 and Table 3).

#### 3.1.3. The Impact of Scaffold and 3D Setup Compared to 2D Cultures on the Genetic Profile of OvCa Cells

We explored the transcriptional landscape of 2D and 3D cultures using three different scaffolds (agarose, collagen, and Matrigel) for the OVCAR8 cell lines.

The cells grown in 3D on Matrigel and agarose and those grown on a basement layer of normal omental fibroblasts embedded in collagen were compared with standard 2D monolayer cultures (Figure 5). The expression profiles of the top 150 DEGs with respect to the growth conditions are shown in Figure 5A (Appendix A). This gene set shows a large variability across the four growth conditions. Initial observations revealed a high degree of similarity in the gene expression between the samples grown in agarose and Matrigel. The collagen samples, however, show an expression profile that diverges from the 2D expression profile to a lesser extent than the OVCAR8 grown on other scaffolds. T-SNE analysis (Figure 5B) recapitulated these observations, showing a partial clustering of the 3D profiles, with the collagen 3D cultures standing out and showing the highest level of similarity to the 2D culture experiments. The top functional groups of the differentially regulated genes included key metabolic pathways, such as glycolysis (Figure 5C).

Next, we explored the genes’ transcriptional signatures in the three scaffolds and the 2D control experiments. We clustered the genes based on their pairwise co-expression scores and visualised them using a uniform manifold approximation and projection dimensionality reduction technique (UMAP) (see Figure 6A). We found localised phenotypic clustering patterns in the OvCa embedded in collagen and agarose, with less variance in the phenotypic expression recorded for the samples grown in Matrigel when compared to 2D. Moreover, the Matrigel cultures showed an inverted gene expression signature compared to the 2D control experiments. Similarly, we analysed the cancer hallmark sets with the DEGs of OVCAR8 grown in 2D compared to 3D data (see Figure 6B). Processes with high covariance included: K-Ras signalling, angiogenesis, interferon alpha and gamma response, TNF alpha signalling, and epithelial-mesenchymal signalling.

#### 3.1.4. Functional Enrichment-2D vs. 3D

A panel of genes was identified as commonly dysregulated in 3D cultures compared to 2D growth conditions. The cumulative 3D data encompassed OVCAR8 grown on Matrigel, agarose, and collagen, while the control data were composed of the experiments using 2D growth conditions. The following genes showed statistically significant differential expression (*p* < 0.05): *C3*, *CXCL1*, *CXCL8*, *IL1B*, *SLPI*, *FN1*, *IL6*, *DDIT4*, *PI3*, *LAMC2*, *CCL20*, *MMP1*, *IFI27*, *CFB*, *ANGPTL4*, and *CXCL2* (Figure 7). Furthermore, gene set enrichment analysis revealed that when grown in 3D, many processes associated with the hallmarks of cancer were also differentially regulated (Appendix A). Key processes that often show enhanced presentation in 3D growth, such as angiogenesis, apoptosis, and hypoxia, all exhibited enrichment as well.

#### 3.1.5. Scaffold-Specific Biomarkers-2D vs. 3D

Next, we examined the transcriptional landscape to identify the potential biomarkers of growth conditions (Figure 8). For this, we used a variety of machine learning algorithms as implemented in OmicsPlayground v2.8.10 to compute a cumulative importance score for all DEGs. The results highlight eight key genes that can be used as predictive scaffold biomarkers (Figure 8A). Specifically, the cells grown in agarose showed condition-specific expressions of four genes: *C3*, *MMP1*, *IL1B*, and *CCL20*. Three potential markers of the cells grown in collagen were identified, namely the interferons *IFI44L* and *IFI27,* and *COL3A1*. Matrigel was represented with only one significant growth marker: *DDIT4*. While these eight biomarker candidates showed the highest importance scores, a variety of other genes showed scaffold-specific expression as well (Figure 8H), suggesting that a number of gene panels can be created to evaluate the impact of growth conditions on the genomic biology.

#### 3.1.6. Cell Line Specificity Impact on Scaffold Selection

Following the analysis of the impact of the scaffold and the 3D vs. 2D environment on the transcriptional landscape of ovarian cancer cell lines, we looked at the differential expression patterns among various cell lines grown on agarose and collagen scaffolds. As expected, we found a good separation of the cell line gene expression characteristics on both scaffolds (Figure 9A,B) using the top 150 differentially expressed genes. Most cell lines also showed fair discrimination between the 2D and 3D cultures on agarose and good segregation among the cancer subtypes (Figure 9C). However, A1847, OVCAR3, OVCAR4, and SKOV-3 on agarose and all cells on collagen (Kuramochi, OVCAR4, and OVCAR8) showed poor differentiation between the growth conditions, suggesting that these scaffolds are potentially not optimal for recapitulating the tumour environment more accurately than the classical 2D cultures in these cell lines.

Functional analysis reflected the diversity of the cell lines grown on each scaffold (Figure 9D). With sex hormones, specific pathways characterised the agarose cultures, while cell growth, development pathways, and fatty acid metabolism were the dominant features of the collagen-grown cell lines. The scaffold impact on cell-line specificity was explored by comparing the differentially expressed genes between OVCAR4 and OVCAR8 in agarose and collagen (Figure 9E). We found that there was a good level of correlation between the gene expression fold changes of the two cell lines for agarose and collagen. Of the top differentially expressed genes, three, *SLC34A2*, *LY6K*, and *BMP7*, showed the same level of dysregulation between OVCAR8 and OVCAR4 in both growth conditions. However, we also identified 13 genes that showed a scaffold-specific differential expression pattern between the two cell lines: *MMP7*, *LAMA3*, *IGFL1*, *S100A14*, *ELF3*, *CYGB*, *ITGB6*, *DKK1*, *TACSTD2*, *IL7R*, *LGALS13*, *IFI6*, and *FOXD1* were collagen-specific, and *IL1B*, *MMP1*, *CP*, *UBB*, *NUPR1*, *SCGB2A1*, *GPNMB*, *IGFBP2*, *GDF15*, *CCL20*, *CYP1A1*, *VTCN1*, and *KRT19* were agarose-specific.

Finally, the differential expression patterns identified a set of genes that showed a cell, a tumour subtype, and scaffold-specific behaviour and can be used as growth environment biomarkers (Figure 9F–H).

#### 3.1.7. Recapitulation of 3D OvCa Using GelTrex

Leveraging the lessons learned from the study of the transcriptional landscape of OvCa cell lines in different conditions, we attempted to capture the phenotypic changes in vitro between the 2D and 3D cultures. For this, we grew SKOV-3 cells in 3D using the hydrogel-based scaffold GelTrex^TM^. Hydrogel was chosen as it encompasses one of the most common scaffolds in the literature, and it is not animal-derived. In addition, this work sought to assess the ease of using a non-established methodology for in-house recapitulation. As such, hanging drop and ultra-low-attachment plates were not included, as their use with OvCa is well established in the literature.

Figure 10 shows the growth of cells over the course of a 9-day period. Here, we adopted a simplistic approach and used a previously tried and tested gel known as GelTrex^TM^. Following the embedding process, the cells began to aggregate and form spheroid-like structures [27]. These structures maintained their circularity and continued to expand in volume as time progressed.

## 4. Discussion

As OvCa is one of the most lethal gynaecological malignancies, there is a clear need for robust models that will help uncover the molecular mechanisms underpinning the disease development, growth, metastasis, and even potential therapeutic responses. Cancer modelling over the decades has progressed from crude anatomy to in vitro cultures, in vivo animal models, and now, in vitro 3D cultures capable of recapitulating in vivo systems and the associated TME. In this meta-analysis, we examined the impact of various scaffolds on the transcriptomic landscape of ovarian cancer cell lines, as well as the differences arising from 3D cultures compared to the classical 2D approaches.

The initial literature survey pointed out the USA as the spearhead of 3D culture research in cancer, covering over 50% of the published output in the field. Similar to what has been observed in 2D cultures, immortalised cell lines take the forefront with SKOV-3 as the most frequently used option, while primary patient samples are used at a reduced rate. Additional cell lines used include OVCAR3, A2780, PEO1, and OVCAR8. The cell line distribution highlights a strong bias towards White European Ancestry. The percentage of East Asian 3D models in the literature is even lower despite the associations with early disease onset in Asian women [28], recapitulating the need for engaging ethnic populations in cancer research.

Further analysis showed that the associated subtypes of the cell lines used align closely with the trend seen in the actual global incidence rates of OvCa subtypes. HGSOC is the most frequent of the epithelial OvCa subtypes, encapsulating 70% of global cases [29], and making this subtype a prime dataset to study in this work assessing the variability in 3D cultures with respect to classical 2D experiments. It must be noted, though, that in vitro work requires long-term investment, with relevant models, especially in OvCa, a commodity. With the advancement of tissue culture techniques towards more physiologically relevant systems, however, researchers must strive to use validated and up-to-date cell lines or note their limitations in disease modelling to maintain reliable and repeatable data.

In this study, we also demonstrated how scaffolds recapitulate the ECM necessary for cell differentiation and the growth of 3D structures [23]. In OvCa modelling, where a 2D counterpart has been used for comparison, the most frequent scaffolds utilised by researchers are Matrigel, hanging drop, low-attachment plates, and hydrogel.

Hanging drop is particularly useful for assessing diffusion gradients in an accessible format [30]. In terms of OvCa, this method has been utilised in toxicity screening assays for monitoring chemoresistance in drugs, such as cisplatin and Niraparib [17,31]. Grown in ultra-low-attachment plates, OvCa cells show altered mitochondrial function through augmented extracellular acidification rates [32]. Re-sensitisation to treatments in cell lines previously thought resistant is also evident using this method, with a number of BRCA wildtype epithelial OvCa cell lines responding to platinum-based therapeutics and showing an increased rate in apoptosis [33]. Cultures such as those arising from ovarian malignancies and grown in Matrigel often maintain histological features, genetic profiles, and intra-tumoral heterogeneity, similar to the in vivo tumour [34]. Matrigel has also proven to be an effective model of early-stage angiogenesis in an array of cancers, including HGSOC [16]. It must be noted that 3D cultures are often chosen to support the principles of replacement, reduction, and refinement (the 3Rs) towards the more ethical use of animals [35,36]. This was further underpinned by our in vitro studies, where we used a non-animal derivative (GelTrex) to grow SKOV-3 cells in 3D. Interestingly, OvCa cell migration, cell communication, and chemotherapeutic response have all been successfully modelled using hydrogel, a plant-based alternative to animal-derivative scaffolds. Here, cultures showed greater similarity to in vivo mouse models and clinical data than those of 2D cultures [37].

Leveraging the data from the Gene Expression Omnibus (GEO) and the Sequence Read Achieve allowed us to create a detailed picture of the genomic landscape of ovarian cancer cell lines in 3D cultures using three distinct scaffolds: Matrigel, agarose, and collagen. All OvCa cell lines showed a high level of differential regulation, with an average of 551 DEGs per dataset ranging from 234 DEGs as the minimum and 1429 DEGs as the maximum. The HGSOC cell line OVCAR8 used across multiple studies allowed us to identify key genes and biological processes that are hallmarks of 3D cultures as well as potential biomarkers of the growth environment for the examined scaffolds. Specifically, our analyses highlighted a set of 8 genes, namely *DDIT4*, *ANGPTLA*, *SELENBP1*, *SULF1*, *GAL3ST1*, *TNFAIP3*, *LLNLR-263F3.1*, and *MUC12*, which showed statistically significant differential expression patterns in the 3D systems compared to 2D, irrespective of the scaffold used. Furthermore, 13 genes showed an environment-specific expression pattern. The top 16 DEGs between 3D and 2D OVCAR8 were also identified. Of note, many of the genes identified are key regulators of inflammation and immune response, such as *C3*, *CXCL8* (*IL-8*), *SLPI*, *CXCL1*, *CXCL2*, *ILI beta*, *IL6*, *CCL20*, *IFI27*, and *CFB* [38,39,40]. Furthermore, many of the top genes also show structural importance in the ECM, e.g., *LAMC2*, *PI3*, *FN1*, and *MMP1*. Dysregulation of the matrix metalloproteinase, *MMP1*, is associated with basement membrane degradation and subsequent peritoneal dissemination in OvCa and is correlated with poor patient prognosis [41]. The remaining DEGs, *DDIT4* and *ANGPTL4*, were recently identified as candidate genes for the prediction of survival outcomes in lung cancer and OvCa patients [42,43].

The functional enrichment scores of OVCAR8 cells grown in Matrigel, agarose, and collagen compared to the standard 2D monolayer controls presented a unique expression profile, with close relations seen among the 2D samples. However, the 3D collagen OVCAR8 cells expressed a higher degree of variability compared to the other 3D OVCAR8 cells, which showed comparatively similar profiles. An earlier study suggested that this model is more similar to the in vivo environment as it captures 3D growth alongside omental fibroblasts [44].

The top biological processes associated with the DEGs identified between the 2D and 3D include glycolysis, KRAS signalling, coagulation, TNF alpha signalling via NF-κB, and complementary and inflammatory response. These processes are frequently altered in cancer and are often difficult to model in 2D systems [21]. Glycolysis, in particular, is often augmented in cancer cells, with increased utilisation of this pathway indicative of the Warburg effect [45]. Similar metabolic changes are also evident in 3D colorectal cancer cells when compared to 2D [22]. The inclusion of these processes in the data verifies numerous studies where 3D cells have been shown to express more biological relevance to in vivo systems than 2D cell cultures through the expression of pathways typically associated with in vivo environments [22,46,47,48].

Furthermore, some cancer-related hallmarks were also highlighted as differentially regulated in the 3D OvCa cells when compared to the 2D samples. Hallmarks of particular interest include apoptosis, oxidative phosphorylation, MYC pathways, ROS, EMT, KRAS signalling, angiogenesis, and hypoxia. Numerous studies have shown that the 3D environment influences these key cancer pathways [22]. Here, we show that regardless of the scaffold, the processes were still heavily influenced when grown in 3D. Apoptosis, EMT, KRAS signalling, hypoxia, and angiogenesis were some of the key cancer-associated processes enhanced in 3D growth. Additional processes included complementary and inflammatory response pathways, which are important factors of tumour immune evasion. Another pathway often seen in cancers is IL6-JAK-STAT3, which is a proliferative driver often implicit in OvCa angiogenesis and tumour metastasis [49].

Moreover, based on the expression profile of the OVCAR8 cells grown in 3D vs. 2D, we identified a panel of genes specific to OVCAR8 when grown in different gel-based scaffolds using the Omics Playground importance score ranking [26]. The expression profile of these genes was unique to the specific scaffold when compared to the 2D OVCAR8. The biomarkers specific to OvCa cells grown in agarose compared to 2D include: *C3*, *MMP1*, *ILIB*, and *CCL20*. The three biomarkers identified for collagen include: *IFI27*, *COL3A1*, and *IFI27*. Matrigel, however, only showed one unique marker, *DDIT4* a stress-included regulator of *mTOR* previously mentioned for its association with progression-free survival in OvCa [43]. Future work should explore the relevance of these markers and the influence they have on the OvCa TME.

Next, we explored the impact of the cell line on various scaffolds and showed that there is a close relationship between the two, suggesting that in order to recover the tissue-specific behaviour in a model 3D culture, a lot of care must be given to the choices of cell line and scaffold in order to remove potential experimental biases. Furthermore, the condition-specific gene expression patterns suggest that a number of genes can be used as environment biomarkers. Finally, we explored the impact of transcriptional changes in real time by looking at phenotypic changes in cells grown in 3D vs. 2D cultures. Our experiments have shown that SKOV-3 cells grown in hydrogel are clustering to form simple spheroids, precursors of higher-order organoid formations. Moreover, similar spheroid formations were also observed to be formed by malignant cells shed from ovarian tumours [50], highlighting the importance of studies on 3D cultures and spheroids in cancer.

Regarding the in vitro study, it should be noted that there was an attempt to recreate the 3D cultures using our own experimental setup, in addition to the published in silico data. To the best of our knowledge, no study has used the SKOV-3 cell line and GelTrex as a scaffold for a 3D cell culture. Therefore, we provide further evidence that a serous ovarian cancer cell line that is mutant for P53 and exhibits ‘BRCAness’ can also form spheroid-like structures using GelTrex. However, there are certain limitations of this study, as it lacked changes in the differentially expressed genes in hydrogel-cultured 3D vs. 2D cells. Future studies should identify any changes in the expression of scaffold-specific genes, EMT, cytokines, or DEGs that have been obtained from in silico studies.

Overall, our research has shown that modelling ovarian cancer is a complex and difficult task, and while 3D cultures have been shown to sometimes more closely reflect the natural environment, our study shows that similar to earlier analyses in other female cancers [51], in ovarian cells, the choice of growth medium can indeed impact the genome function and activity.

## 5. Conclusions

In summary, this meta-analysis assessed the current landscape of 3D OvCa models in the literature and provided a complex expression profile of OvCa cells grown in 3D. Our transversal comparison of various scaffolds allowed us to highlight the variability that can be induced by various scaffolds in the transcriptional landscape and identify key genes and biological processes that are hallmarks of cancer cells grown in 3D cultures. Moreover, the identification of transcriptional signatures that show genes’ specificity in cell lines, tumour subtypes, and scaffolds, and are defined as growth environment biomarkers will allow us to monitor, in the future, the suitability of 3D cultures to recapitulate tissue complexity.

## Figures and Tables

**Figure 1 cancers-15-03350-f001:**
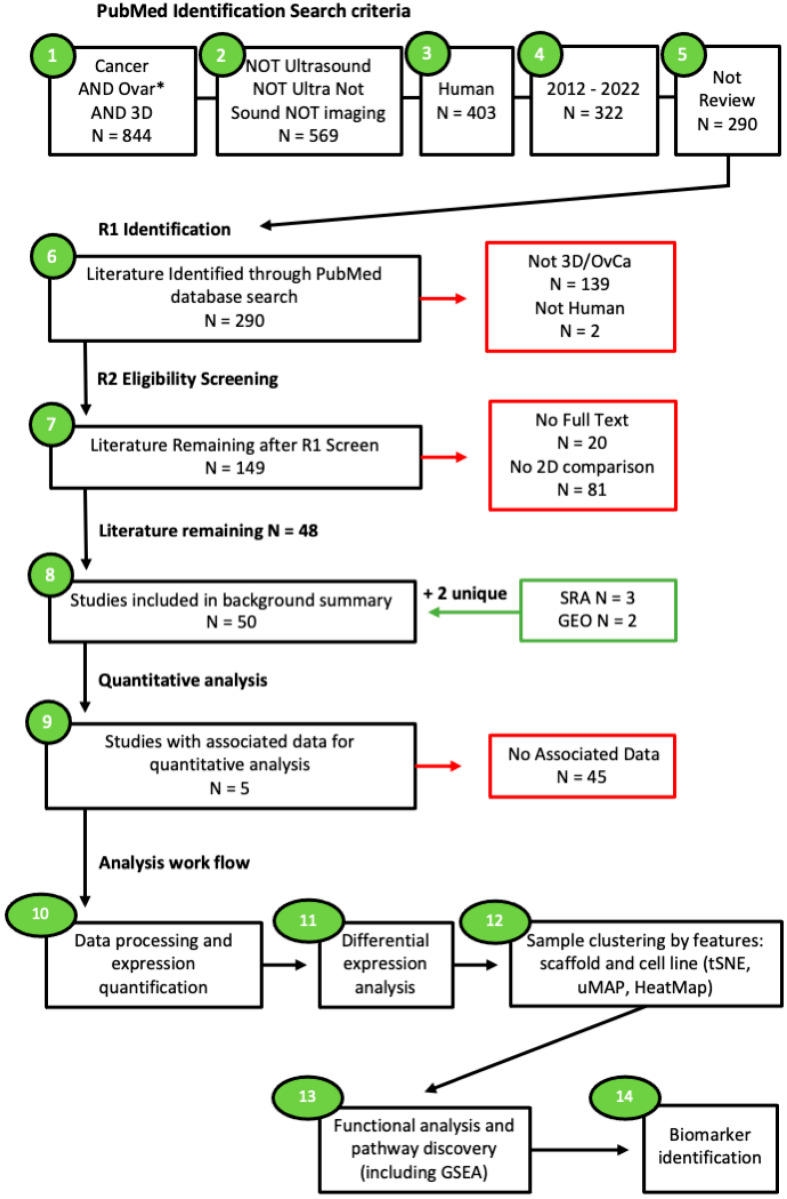
Search criteria workflow. Studies accessed through Pubmed.gov on 25 June 2022. Using pre-defined search terms. Articles were subjected to two rounds of screening by two independent reviewers. Additional data sought through Sequence Read Archive and Gene Expression Omnibus on 1 July 2022. Studies were split into two groups: those suitable for the background summary (N = 50) and those containing associated data (N = 5). Studies with associated data were then processed prior to functional enrichment analysis and biomarker identification. The “*” used in the search criteria is a wild card allowing for all words matching the associated string to be retrieved.

**Figure 2 cancers-15-03350-f002:**
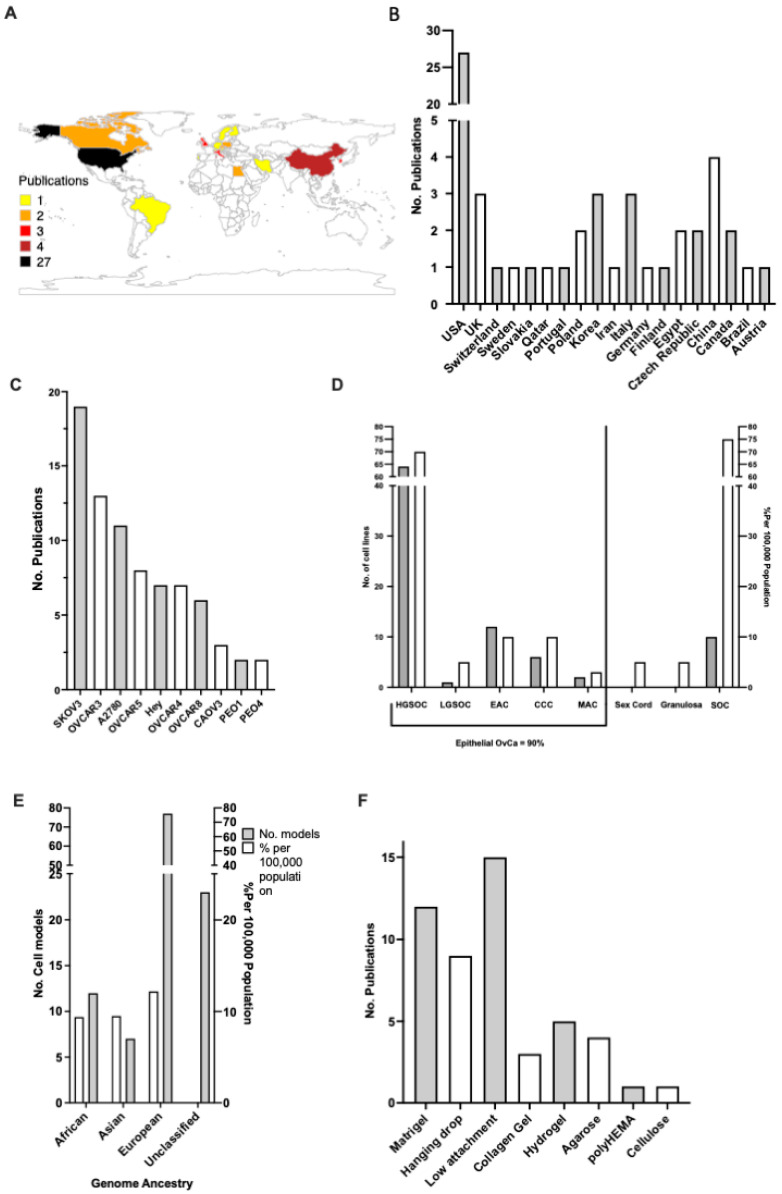
Overview of the published 3D culture experiments in ovarian cancer. (**A**) Gradient map depicting the global spread of publications 2012–2022. (**B**) Chart showing no. of publications per country 2012–2022. (**C**) Top cell lines used for 3D ovarian cancer (OvCa) in the literature. (**D**) Trends between the distribution of cell models against actual global rates (white) pertaining to OvCa subtype (grey). (**E**) Genome ancestry of cell lines used (grey), contrasted with actual global OvCa ethnicity rates (white) (2012–2022). (**F**) The ten most frequently used scaffolds for supporting the growth of OvCa cells (circa 2012–2022) selected from the publication corpus analysed.

**Figure 3 cancers-15-03350-f003:**
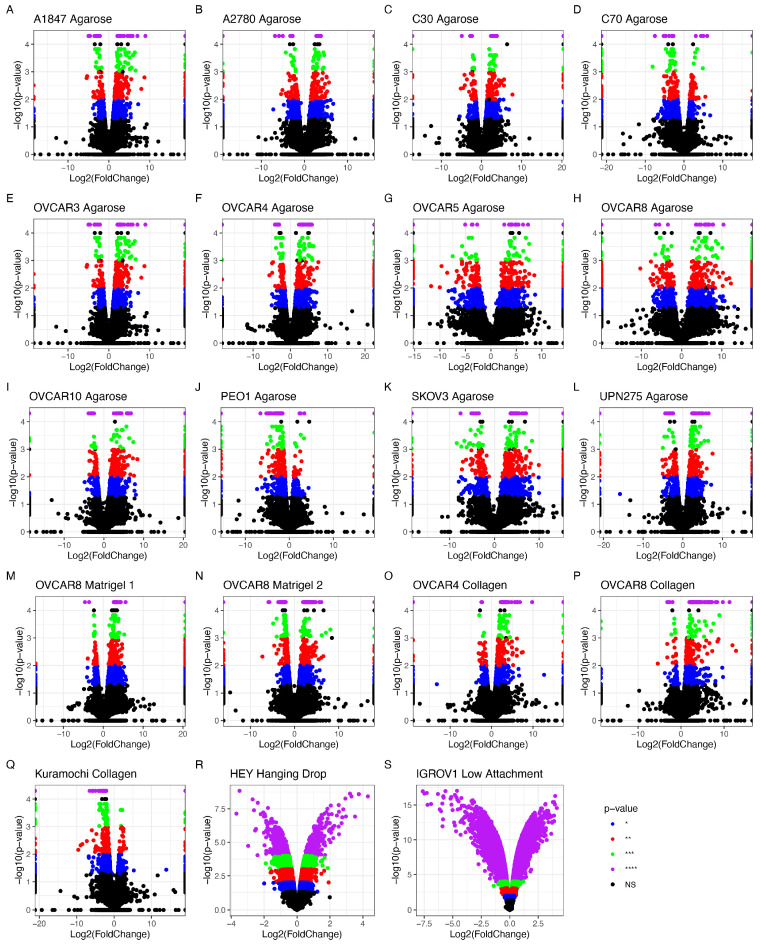
Differentially expressed genes (DEGs) detected by RNA sequencing analysis of OvCa cell lines grown in 2D compared to 3D. (**A**–**Q**) show data extracted from RNAseq experiments; (**R**,**S**) show data extracted from microarrays. Significance thresholds for (**A**–**Q**) were set at NS > 0.05 = grey/black, * *p* < 0.05 = blue, ** *p* < 0.01 = red, *** *p* < 0.001 = green, and **** *p* < 0.0001 = purple. (**R**,**S**) *p*-value threshold = 0.05; NS data are shown in black. (**A**–**L**) have agarose as scaffold, (**M**,**N**) are Matrigel, (**O**–**Q**) are collagen, (**R**) is hanging drop, and (**S**) is low-attachment. (**A**) A1847 endometrioid carcinoma of the ovary (EAC); (**B**) A2780 EAC; (**C**) C30 carcinoma; (**D**) C70 carcinoma; (**E**) OVCAR3 HGSOC; (**F**) OVCAR4 HGSOC; (**G**) OVCAR5 HGSOC; (**H**) OVCAR8 HGSOC; (**I**) OVCAR10 HGSOC; (**J**) PEO1 HGSOC; (**K**) SKOV-3 carcinoma; (**L**) UPN275 mucinous adenocarcinoma (MAC); (**M**) Kuramochi HGSOC; (**N**) OVCAR4 collagen HGSOC; (**O**) OVCAR8 Matrigel 1 HGSOC; (**P**) OVCAR8 Matrigel 2 HGSOC; (**Q**) OVCAR8 collagen HGSOC. (**R**) HEY HGSOC; (**S**) IGROV1 EAC.

**Figure 4 cancers-15-03350-f004:**
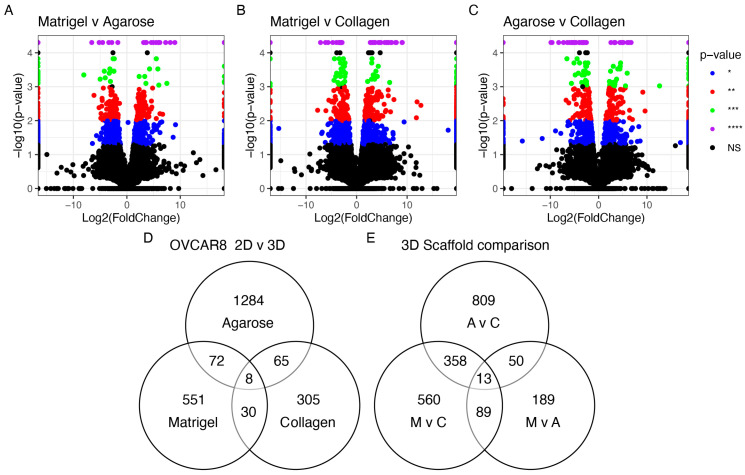
Differentially expressed genes seen in OVCAR8 grown in 3D. (**A**) Agarose vs. collagen; (**B**) Matrigel vs. agarose; (**C**) Matrigel vs. collagen. Threshold set at *p* < 0.05. (**D**) Common genes differentially expressed in OVCAR8 grown in 3D vs. 2D. (**E**) Common genes among (**A**–**C**); M: Matrigel, C: collagen, A: agarose. * *p* < 0.05 = blue, ** *p* < 0.01 = red, *** *p* < 0.001 = green, and **** *p* < 0.0001 = purple. NS > 0.05 = grey/black.

**Figure 5 cancers-15-03350-f005:**
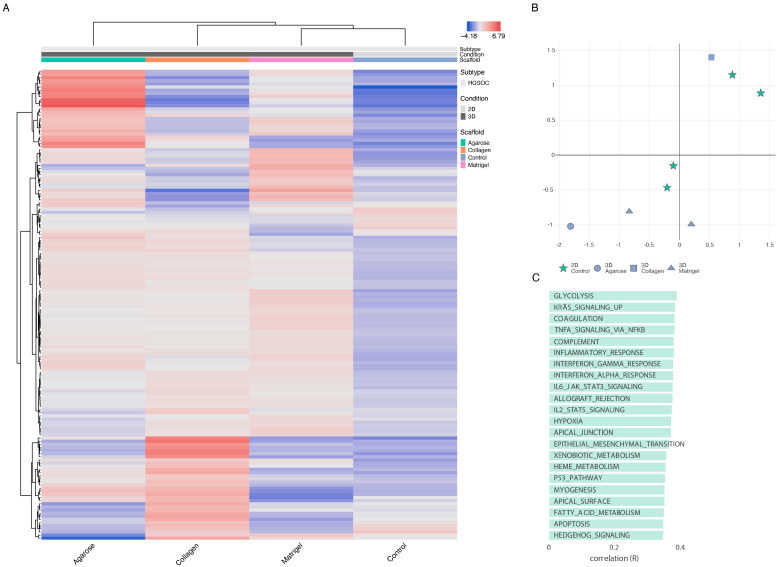
OVCAR8 transcriptional profile in 2D vs. 3D. (**A**) Top 150 differentially regulated genes from OVCAR8 grown under 2D and 3D conditions. Data originating from 3 unique studies, encompassing 4 growth conditions. Three-dimensional cells grown in Matrigel, collagen, and agarose. Two-dimensional cells grown under standard lab conditions as matched controls to each 3D experiment. The gene name list is available in Appendix A. (**B**) T-distributed stochastic neighbour embedding (T-SNE) plot of the genetic profiles of the HGSOC OVCAR8 grown in Matrigel (at 7 and 14 days-triangle), collagen (square), agarose (circle), and monolayer (stars). (**C**) Functional analysis of the top 150 differentially regulated genes between 2D and 3D growth conditions, showing key biological pathways associated with them.

**Figure 6 cancers-15-03350-f006:**
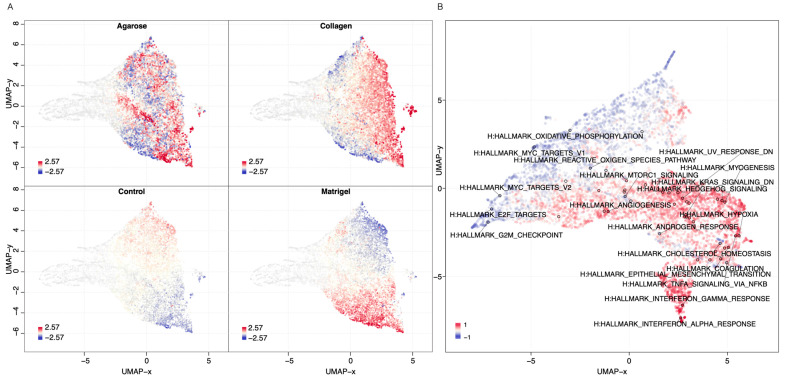
Gene and phenotypic hallmark signature profiles. (**A**) UMAP clustering of genes coloured by relative log-expression in four growth conditions: agarose, collagen, Matrigel, and 2D controls. The distance metric is covariance. Genes that are clustered nearby have high covariance. (**B**) UMAP hallmark covariance using OVCAR8 grown in 2D and combined 3D data. Clustering of associated hallmarks. Processes upregulated in 3D are indicated in red. Downregulated are indicated in blue.

**Figure 7 cancers-15-03350-f007:**
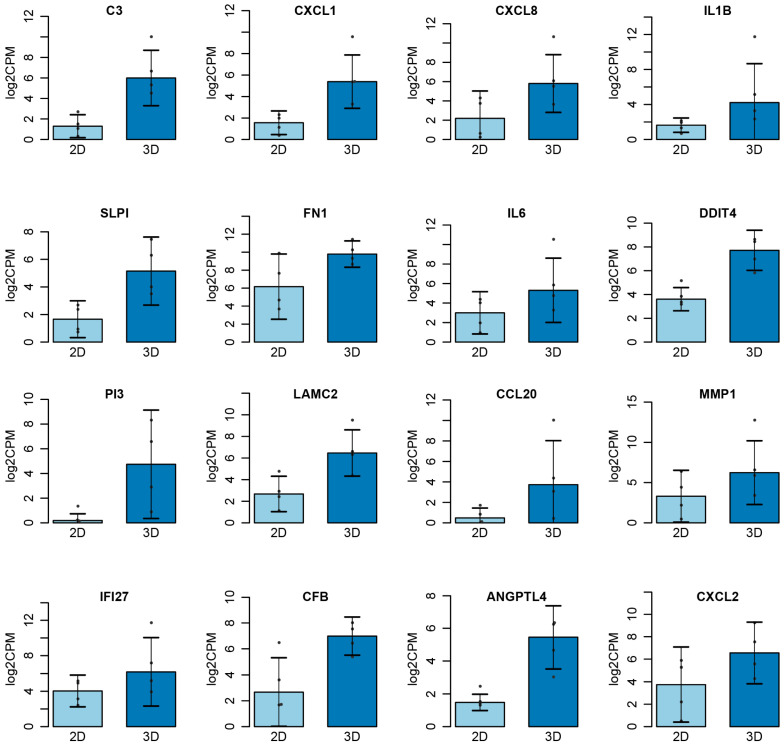
Top genes differentially expressed in 2D vs. 3D. Cumulative data for 3D taken from OVCAR8 embedded in Matrigel, agarose, and collagen. Significance threshold *p* < 0.05.

**Figure 8 cancers-15-03350-f008:**
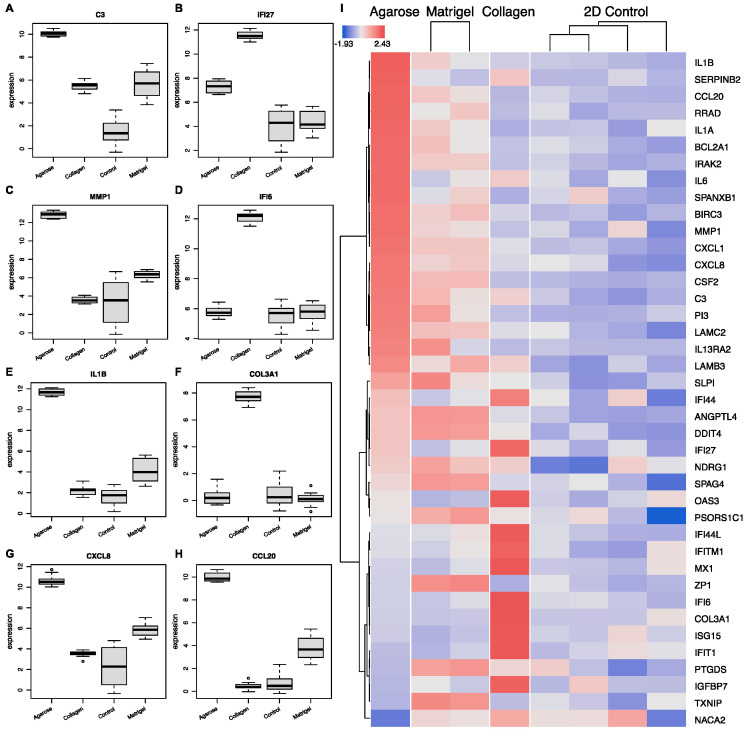
Scaffold-specific biomarker identification. (**A**–**H**) The top 8 genes implicated with expression-specific profiling for each condition. (**I**) Biomarker heatmap: expression heatmap of top gene features according to their variable importance score. Importance scores were calculated based on multiple machine learning algorithms, including LASSO, elastic nets, random forests, and extreme gradient boosting.

**Figure 9 cancers-15-03350-f009:**
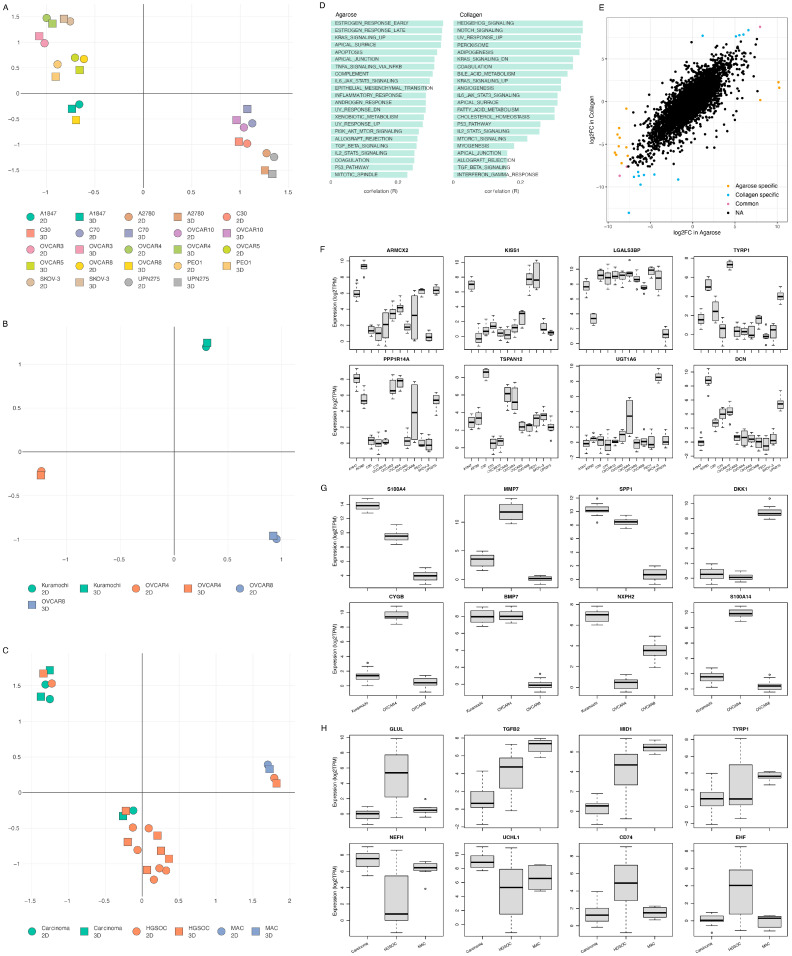
Cell-line-specific transcription in agarose and collagen. (**A**,**B**) T-SNE plot of the genetic profiles of cell lines grown in agarose and collagen, respectively, against a 2D control. (**C**) Umap plot of the transcriptional profile of cancer subtypes in agarose vs. 2D control. (**D**) Functional analysis of the top 150 differentially regulated genes between 2D and 3D growth conditions, showing key biological pathways associated with them for agarose and collagen; the list of genes is shown in Appendix A. (**E**) Similarity of gene differential expression in OVCAR4 vs. OVCAR8 in collagen vs. agarose. (**F**–**H**) The top 8 environment biomarkers for cell lines grown in agarose (**F**,**H**) and collagen (**G**).

**Figure 10 cancers-15-03350-f010:**
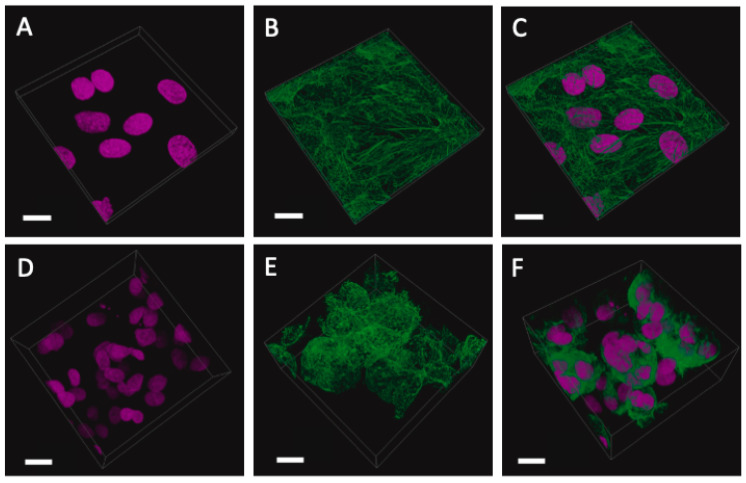
SKOV-3 cells grown for 9 days in conventional monolayer formation compared with those embedded in GelTrex^TM^. (**A**–**C**) Monolayer cells: nuclei (pink), phalloidin (green), and overlay showing a single plane of cells across a flat glass substrate. (**D**–**F**) Three-dimensional cells: nuclei (pink), phalloidin (green), and overlay showing aggregated spheroids with multiple nuclei. The scale bar is 20 μm. Daily snapshots of the growth of SKOV-3 cells from day 2 to day 9 are shown in Appendix A.

**Table 1 cancers-15-03350-t001:** Differences between 2D and 3D cell culture systems.

2D Cultures	3D Cultures
Cells grown in monolayers-biologically simple	Cells form differentiated aggregates, spheroids, or organoids-biologically complex
Gene and protein expression differ from in vivo	Expression closely mimics in vivo
Uniform exposure to chemical stimuli; drugs often appear affective	Nonuniform growth results in toxicity profiles and diffusion gradients closely related to in vivo
Oxygen diffusion is uniform and higher than many in vivo structures, thus augmenting mitochondrial function and ROS production	Oxygen distribution varies and hypoxic cores are evident, closely mimicking in vivo variations of many complexes
Long-term cultures can result in genetic drift, with epigenetic and morphological changes evident	Growth is typically short-term, minimising genetic drift
Can be cheaper and less complex, and therefore, easily recapitulated in a lab	Requires additional nutrients and biological scaffolds, and can therefore be more expensive and time-consuming
Established protocols	Limited established protocols

**Table 2 cancers-15-03350-t002:** Accession codes of RNA sequencing of 2D and 3D OvCa cell models.

Accession	Platform	Paired Reads
PRJNA472611	Illumina HiSeq 2500	24
PRJNA530150	Illumina NextSeq 500	32
PRJNA564843	Illumina NextSeq 500	36

**Table 3 cancers-15-03350-t003:** OVCAR8 genes commonly differentially regulated in 3D conditions grown on agarose, collagen, and Matrigel compared to 2D cultures.

**Common** **3D vs. 2D**	**Datasets**	**Scaffold** **Specific**	**Datasets**
*DDIT4*	12	*RP11-13K12.2*	0
*ANGPTL4*	15	*EEF1A1P9*	0
*SELENBP1*	7	*EEF1A1P12*	0
*SULF1*	6	*TENM2*	5
*GAL3ST1*	7	*RP11-297P16.4*	3
*TNFAIP3*	9	*GGT1*	1
*LLNLR-263F3.1*	4	*IFI44*	5
*MUC12*	4	*CXCL2*	3
		*KIF1A*	2
		*AC003092.1*	3
		*INHBA*	6
		*RP13-143G15.4*	7
		*GREM1*	3

## Data Availability

RNAseq and array data can be found via the following NCBI accession codes: PRJNA472611, PRJNA530150, PRJNA564843, PRJNA564843, PRJNA232817, and PRJNA318768. A full list of samples can be viewed in Appendix A.

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
