# Peer review of "Transcriptional Landscape of 3D vs. 2D Ovarian Cancer Cell Models"

_cancers, 2023, doi:10.3390/cancers15133350_

Round 1
Reviewer 1 Report
The manuscript by Kerslake et al focusses on the importance of studying 3D culture than 2D cultured ovarian cancer cell lines and evaluated the variability induced by different scaffolds on the transcriptional landscape of 3D vs 2D cultures. The authors have also identified key genes and biological processes that are hallmarks of cancer cells grown in 3D cultures. The compilation of data across several studies combined with transcriptional analysis is interesting. However, few minor points need to be addressed for the improvement of this study.
1. Title: Since the authors have analyzed the dataset along with compilation of previous literature, the title can be changed to “Transcriptional landscape of 3D vs 2D ovarian cancer cell models”
2. Tables: Please revise supplementary S2 table to make it more understandable to the readers. Some of the cell line data seems to be missing as per the table legends. Please revise accordingly
3. Figure: Fig S1 figure legend needs to be improvised. Do these GSEA belong to 3D spheroids? If yes, then the legends can be simply written as GSEA of 3D spheroids instead of 2D vs 3D which is a bit confusing to the readers. Font size of all figures need to be increased.
4. Figure 10: Since OvCa cell migration, cell communication, and chemotherapeutic response have all been successfully modelled using hydrogel, Fig 10 is simply a repetition of previous studies. The statement made by authors “The results suggest that the changes at genomic level have a direct impact on the 3D aggregation of cells” remains inconclusive. The authors have already compared the effect of agarose, collagen and Matrigel on transcriptional landscape. Therefore, it would be interesting to see the changes in differentially expressed genes in hydrogel cultured 3D vs 2D cells. as well. In Fig 10, along with the Z-stacked confocal and phase contrast IF images (not in 3D plane) for representation purpose only, please include the expression levels of key genes such as scaffold specific genes, EMT, cytokines, genes from Fig 7 (obtained from RNA sequencing and Microarray datasets) in day10 3D spheroids vs 2D using qPCR or qPCR array.
5. The authors are suggested to mention a brief flowchart of the whole study including the meta-analysis, transcriptomic analysis and IFs.
6. RNA-Sequencing analysis: Since omental fibroblast can itself change the transcriptional profiling of cancer cells; the differential gene expression obtained cannot directly reflect the effect of scaffold-Collagen used in this study. Do the authors have RNA seq datasets of 3D spheroids grown only on collagen which is more appropriate? What is the FDR used for RNA seq analysis?
7. Microarray analysis: What is the FDR, pvalue, Log2FC used for microarray (GEO datasets). Please mention in GEO2R analysis that they are microarray datasets from Affymetrix platform.
8. Methods: What is the cell number used for 3D spheroids? How did authors make sure they are 3D spheroids and not simply cell aggregates? Please include the phase contrast images of 2D and 3D from Day 1 to 10 (not all, atleast 3-4) along with IF images captured in Z-stacking. Phase contrast images need not be in Z-Stacking.
9. Discussion: Authors are suggested to refer to some of the recent publications (PMID: 33087324; PMID: 35205706, PMID: 35626175) and include the importance of studying 3D spheroids in ovarian cancer in the context of peritoneal spread of the disease in discussion. In this study (PMID: 33087324) authors have discussed the mode of peritoneal spread where malignant cells are often shed into the peritoneal fluid as nonadherent form (3-D spheroids) where they survive as spheroid- like aggregates, which later spread through the peritoneal fluid to abdominal organs, and then attach and grow as adherent colonies (2-D). Additionally, an interesting and recently published study (PMID: 35205706, PMID: 35626175) showed that some cancer cells when seeded in as monolayer can form spontaneous floating viable spheroids. Such spontaneous forming spheroids are characteristics of aggressive phenotype that are prone to chemoresistance. This intrinsic ability of cancer cells to form spontaneous spheroids is an excellent tool to identify appropriate regiments for personalized therapy for ovarian cancer.
10. References: Ref 6 has been mentioned twice in ref 51, Ref 21 and 47 are in repetition. Many of the references have been mentioned twice. Please rearrange and remove the duplicates.
Author Response
Dear Reviewer,
We thank you for your comments and we have addressed them in a point wise manner in the attached document.
Kind regards,
Cristina Sisu

Reviewer 2 Report
Title: “A Meta-analysis of 2D vs. 3D Ovarian Cancer Cellular Models” Authors: Rachel Kerslake, Birhanu Belay, Suzana Paniflov, Marcia Hall,Ioannis Kyrou, Harpal Randeva, Jari Hyttinen, Emmanouil Karteris,
Cristina Sisu
Summary:
This meta-analysis addresses an interesting topic that may enrich the design of in vitro scientific work on ovarian cancer in the future.
Several points are listed below:
1: It appears from the abstract that 2D models from 2012-2022 were studied. From which years are the 3D models compared to this? The text passage is misleadingly worded in comparison to material and methods.
2: In the introduction, the authors mention that the 3D models are also tested on cells of other tumor types. Here it would be worth mentioning that 3D models in breast cancer cells and CRC cells have now even been extended to multicellular 3D cultures: Example references:
doi 10.3389/fonc.2021.764204,
doi 10.3389/fimmu.2022.904418,
doi 10.3389/fphar.2021.699842,
doi 10.3390/ijms23094714.
3: Building on this, the Discussion could mention whether multicellular 3D models for OvCa cells would also be conceivable and useful in the future.
4: In Material and Methods it states that all reagents were obtained from Thermofisher. Where did the cells used come from? Please add.
5: For better orientation, it would be helpful if the subchapters were numbered.
6: The quality of some figures (2, 8, 9) could be improved.
Author Response
Dear Reviewer,
We thank you for your comments and we have addressed them in a point wise manner in the attached document, and the revised manuscript and supplementary materials.
Kind regards,
Cristina Sisu

Reviewer 3 Report
This is a valuable meta-analysis, but some corrections are mandatory.
1. There are noticeable deficiencies in the preparation of the paper by the requirements of the journal.
2. each abbreviation in the abstract and main text has to explain; please give the full name.
3. the aim of this paper does not indicate that it is a meta-analysis
4. line 43 please delete.
5. the title should be more informative.
6. LogFC2 should be log2FC, where 2 is in subscript
7. abbreviation of genes please write ithalic
8. However authors tried to prepare the paper as good as possible I still feel a hunger for information. I think the literature should be better, more thoroughly researched and described.
9. please indicate strengths and weaknesses
10. please remember each time that this is a meta-analysis
11. please provide future directions.
12. Some typo mistakes should be eliminated.
Author Response

(The authors gave the same response as above.)
